# Drug and Protein Interaction Network Construction for Drug Repurposing in Alzheimer's Disease

**Georgios N. Dimitrakopoulos** , **Aristidis G. Vrahatis, Themis P. Exarchos, Marios G. Krokidis** *
and **Panagiotis Vlamos**

Bioinformatics and Human Electrophysiology Laboratory, Department of Informatics, Ionian University,
49100 Corfu, Greece; dimitrakopoulos@ionio.gr (G.N.D.); aris.vrahatis@ionio.gr (A.G.V.);
exarchos@ionio.gr (T.P.E.); vlamos@ionio.gr (P.V.)
* Correspondence: mkrokidis@ionio.gr

**Abstract:** Alzheimer's disease is one of the leading causes of death globally, significantly impacting countless families and communities. In parallel, recent advancements in molecular biology and network approaches, guided by the Network Medicine perspective, offer promising outcomes for Alzheimer's disease research and treatment. In this study, we aim to discover candidate therapies for AD through drug repurposing. We combined a protein-protein interaction (PPI) network with drug-target interactions. Experimentally validated PPI data were collected from the PICKLE meta-database, while drugs and their protein targets were sourced from the DrugBank database. Then, based on RNA-Seq data, we first assigned weights to edges to indicate co-expression, and secondly, estimated differential gene expression to select a subset of genes potentially related to the disease. Finally, small subgraphs (modules) were extracted from the graph, centered on the genes of interest. The analysis revealed that even if there is no drug targeting several genes of interest directly, an existing drug might target a neighboring node, thus indirectly affecting the aforementioned genes. Our approach offers a promising method for treating various diseases by repurposing existing drugs, thereby reducing the cost and time of experimental procedures and paving the way for more precise Network Medicine strategies.

**Keywords:** Alzheimer's disease; network medicine; drug repurposing; RNA-sequencing; protein-protein interaction

## 1. Introduction

Alzheimer's Disease (AD) stands as one of the most daunting challenges in modern medicine [1]. As the most common form of dementia, AD affects millions worldwide, with numbers projected to rise as the global population ages. The complexity of AD's etiology, combined with the limited efficacy of current treatments, underscores the urgent need for innovative therapeutic strategies. Traditional drug discovery methods, while invaluable, often entail prolonged timelines and exorbitant costs [2]. In light of these challenges, there's a growing consensus in the scientific community about the potential of alternative approaches, such as systems biology and network medicine, to offer fresh insights into the disease's intricacies [3].

In parallel, the concept of drug repurposing also known as drug repurposing, has gained traction [4]. Instead of designing new molecules from scratch, drug repurposing seeks to identify new therapeutic applications for existing drugs. While this endeavor is complex, it holds immense promise in decreasing drug development costs and enhancing safety [5]. The foundation of drug repositioning lies in the understanding that a single drug can influence multiple targets, that distinct diseases might share cellular and molecular traits, and that a single target can have varied effects. Modern high-throughput technologies are producing data at an unprecedented rate, promoting the use of computational methods to discern links between drugs, diseases, and targets, thereby enhancing the drug

repurposing journey [6]. Current data analysis techniques encompass statistical methods, machine learning, and notably, models based on biological networks. These digital tools are bridging the gap between data production and its interpretation in the biomedical domain [7].

In the implementation of search strategies for drug repurposing, drug-protein interaction networks, which map the relationships between drugs and their protein targets, play a pivotal role [8]. For AD, these networks can unveil potential drug candidates that might modulate key pathways implicated in the disease [9]. This approach not only accelerates the drug development process but also reduces associated risks and costs. Researchers can save time using already-approved drugs, as they can perform direct phase II clinical trials. Several promising compounds are reported that are prioritized to be used in clinical trials with AD patients [10]. For example, several interesting studies in the field of metallodrugs consider anticancer cisplatin derivative as a modulator of amyloid aggregation [11,12]. Yet, as with network medicine, the expanding pool of data underscores the need for more advanced tools to harness the full potential of drug-protein networks.

Emerging strategies, like network-based drug-disease proximity, illuminate the connections between drugs and their disease-related molecular targets. These methods offer a streamlined way to identify new uses for approved drugs known for their safety, efficacy, and potential side effects [13]. However, to rank these repurposed drug candidates or propose new treatments, thorough validation is crucial. Since these strategies focus on already-approved drugs, it's feasible to test hypotheses using extensive patient data from regular healthcare practices. This real-world data complements evidence from randomized controlled trials, which, despite leading to drug approval, often have limitations due to their smaller sample sizes, shorter durations, and lack of diverse patient representation.

In this work, our aim is to detect potential drug targets for AD based on known drug-proteins and protein-protein interactions (PPIs). Focusing on proteins that play a significant role in AD but without known drugs, the key idea is to detect drugable neighbors of these proteins. Towards this aim, we created a drug-protein graph from high-confidence PPI and drug-protein interaction data. Then, a scRNASeq dataset on AD was used as input to a classification method in order to detect genes whose expression presents different patterns between AD and control conditions. Then, for the corresponding proteins of the top-ranked genes, we searched in the graph if they were targeted by a drug or not. In the latter case, we searched for drugs targeting another protein that is linked with a direct PPI. Further analyzing these drugs, we found that in several cases, although their primary aim is not AD, they have the potential to be repurposed for this disease.

## 2. Materials and Methods

In summary, we collect high-fidelity data for protein-protein interactions as well as drug-protein interactions to construct a single graph containing both types of interactions. Then, a recent scRNASeq dataset on AD is used in conjunction with machine learning algorithms to detect genes that differentiate between disease and control conditions. Enrichment analysis confirmed that these genes are involved in processes related to AD or other neurodegenerative diseases. Finally, for the genes/proteins of interest, we search if they are drug targets or if some neighbor of them has known drug targets and we further analyze some indicative cases.

### 2.1. Graph Data

The PPI interaction network of the human was meticulously downloaded from the Protein InteraCtion KnowLedgebasE (PICKLE, www.pickle.gr, (accessed on 5 August 2023) meta-database [14]. Serving as a remarkable resource, PICKLE functions as a meta-database by meticulously accumulating direct protein-protein interactions from renowned primary databases such as BioGRID, Intact, and HPRD. What sets PICKLE apart is its unique approach of integrating these interactions with an ingenious ontological scheme. This scheme, deeply rooted in genetic information, revolves around the UniProtKB/Swiss-Prot-

reviewed complete proteome of the human (RHCP). The beauty of this approach lies in its ability to seamlessly amalgamate data across various levels without succumbing to any loss stemming from conversions to specific genetic levels like genes, mRNA, or proteins.

In this endeavor, the prevailing version of PICKLE, namely version 3.3, stood as the cornerstone. The network it harbored was acquired through the "cross-checked" filtering mode at the UniProt level, ensuring a meticulously refined dataset. Within the intricate fabric of this PPI network, a staggering count of 218,025 interactions gracefully intertwined among a constellation of 16,420 nodes.

On a parallel front, the drug-protein network took its form within the realm of UniProt (www.uniprot.org, accessed on 5 August 2023), bearing origins traceable back to the comprehensive DrugBank (https://go.drugbank.com/, accessed on 5 August 2023). This bipartite marvel elegantly encompasses 25,707 interactions that seamlessly bridge 6304 distinct drugs with a web of 3137 protein targets. This intricate interplay of molecular relationships casts a profound light on the complexity that underpins biological systems, inviting us to unravel more layers of this fascinating tapestry.

### 2.2. Gene Expression Data

Two single-cell datasets (with codes AD00801 and AD8003) on AD were obtained from the scREAD database [15], which is a publicly available repository collecting scRNA-seq and snRNA-seq datasets. The datasets originated from postmortem human brain tissues demonstrating AD pathology as well as healthy, non-AD samples (GEO accession code: GSE147528 [16]). The procured datasets encompass human brain cells from the superior frontal gyrus region, with one dataset representing healthy control cells and the other representing cells from AD cases, collectively accounting for a total of 66,612 cells (32,901 disease cells and 33,711 control cells).

The two datasets were integrated using SCANORAMA [17], which is a state-of-the-art algorithm for integration/batch effect correction. Developed to address the challenges arising from the amalgamation of disparate datasets, SCANORAMA offers an advanced framework that harmonizes information from multiple sources while effectively mitigating the influences of batch effects—systematic variations introduced during data acquisition across different experimental runs or conditions. These variations can often obscure meaningful biological signals, making robust correction methods like SCANORAMA essential for accurate and reliable analyses. At its core, SCANORAMA employs sophisticated mathematical and statistical techniques to transform and align the datasets, ensuring that data points from distinct sources are placed on a common reference system. This transformation enables meaningful comparisons and analyses, enhancing the utility of the integrated data. One of the key strengths of SCANORAMA lies in its ability to deal with the intricacies of biological data, where each dataset might present unique characteristics and challenges.

The algorithm leverages a variety of techniques, potentially including dimensionality reduction methods, manifold alignment, and other data-driven approaches, to uncover shared patterns and structures among the datasets. These techniques allow SCANORAMA to identify sources of variation that arise not only from biological differences but also from the technical variations introduced by different experimental setups or conditions. Ultimately, SCANORAMA's goal is to provide a unified view of the integrated data that effectively captures the underlying biological information while mitigating the confounding effects of batch variations. By doing so, researchers can confidently perform analyses, such as identifying commonalities, differences, and correlations across the integrated datasets, leading to more accurate and comprehensive biological insights. Taking advantage of these identified anchors, this methodology efficiently rectifies batch effects, providing a cohesive integration of the datasets. Finally, the top 2000 genes displaying highest variability were selected. This was performed by the "Scanpy" Python package [18], which is a toolkit for analyzing single-cell gene expression data capable of efficiently dealing with large data. The corresponding protein of each gene was obtained via the PICKLE ontological scheme.

*2.3. Drug Repurposing via the Drug-Protein Network*

The scRNASeq dataset was used to detect the most significant nodes related to AD and obtain a ranking of their ability to distinguish health and disease conditions. In detail, the Random Forest method was employed to classify the cells as control or disease, based on gene expression, i.e., the genes were considered as features and the cell samples as observations. Random Forest [19] is an ensemble algorithm consisting of a large number of classification trees (here, 100 trees were used). To avoid overfitting to training data, each tree is given a random subset of samples, while at each node, a random subset of variables is tested to define the best split. Then, the variable importance measurement was extracted, a critical aspect of understanding the behavior and predictive power of machine learning models, including Random Forest. Random Forest is an ensemble learning method that constructs multiple decision trees during training and combines their predictions for more accurate and robust results. Results of Random Forest are considered more robust, especially compared to single tree classification due to randomness: (a) each tree receives a random subsample of data and (b) in each node some variables are selected randomly to define the best split. Variable importance measurement in the context of Random Forest refers to assessing the significance of individual input features (here genes) in contributing to the model's overall predictive performance. We used the normalized Gini Importance, which calculates the total reduction in the Gini impurity (a measure of node impurity in decision trees) achieved by a particular feature over all trees in the Random Forest. Higher values of the variable importance show a higher reduction, indicating that the feature is considered more important.

It is important to note that the calculated variable importance values are relative within the context of the model and dataset used. High variable importance suggests that the feature is influential for the model's predictions, but it does not necessarily imply a causal relationship. Variable importance should be interpreted with caution and ideally cross-validated to ensure its robustness. Specifically, at each node, the difference in class impurity is computed before and after the data split. An important variable would lead to a high reduction of impurity, with this having large variable importance values. For this estimation, the out-of-bag samples were used (the samples that were randomly not selected for each tree training).

We aggregated the PPI and the drug-protein data into a single network with two types of nodes. For each important gene, we search on the graph if any drug exists in the direct neighbors. We noted that for several of them, there is no known drug targeting them. However, considering the second degree of neighbors (i.e., the neighbors of direct neighbors), we aim to detect some drugs, which by affecting the neighbor, might affect the gene of interest, diminishing the effect of the disease. A summary of our method is presented in Figure 1. All processing steps were implemented in Python 3.9.3 using in-house custom scripts.

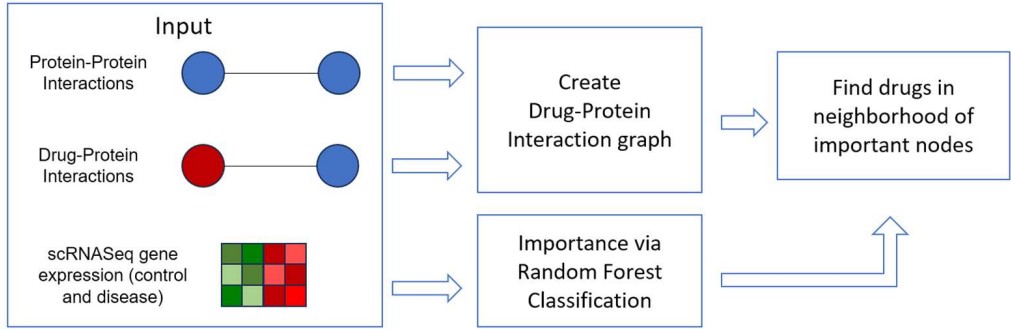

**Figure 1.** Workflow of our methodology.

### 3. Results

#### 3.1. Important Genes

The importance score obtained by Random Forest is depicted in Figure 2. The shape of the score curve is "exponential", i.e., few nodes have a large score and most nodes have a low score. Thus, our data-driven approach is able to isolate a small number of genes that are potentially relevant to AD for further examination. Based on the top 20 most important nodes, we performed enrichment analysis using DAVID [20] to ensure that the detected genes are involved in relevant biological processes. Table 1 summarizes the Gene Ontology terms of Biological Processes that were detected, with the majority of them related to neurodegenerative diseases.

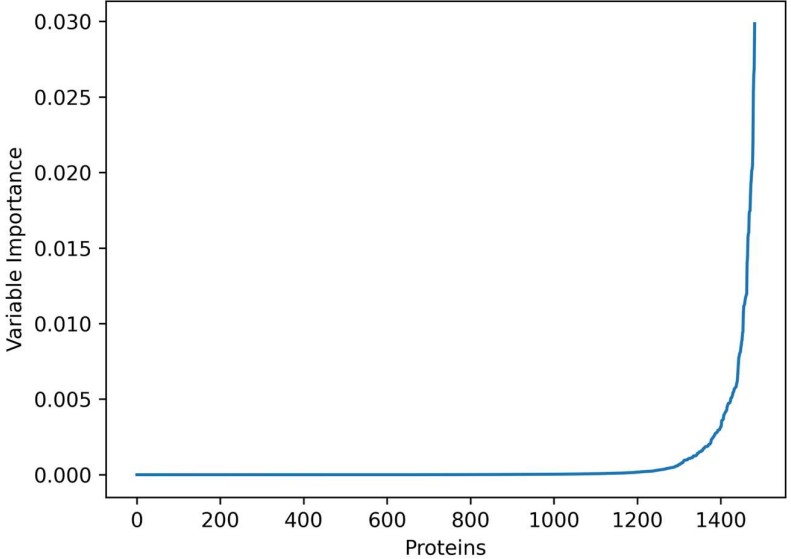

**Figure 2.** Importance of each gene/protein to according to Random Forest classification.

**Table 1.** Gene Ontology Biological Process terms detected in the most important genes.

| Code | Term | $p$-Value |
|---|---|---|
| GO:0014002 | astrocyte development | $1.23 \times 10^{-4}$ |
| GO:2001244 | positive regulation of intrinsic apoptotic signaling pathway | $5.30 \times 10^{-4}$ |
| GO:0070488 | neutrophil aggregation | 0.0019 |
| GO:0032119 | sequestering of zinc ion | 0.0037 |
| GO:0018119 | peptidyl-cysteine S-nitrosylation | 0.0046 |
| GO:0035425 | autocrine signaling | 0.0065 |
| GO:0002544 | chronic inflammatory response | 0.0092 |
| GO:0002523 | leukocyte migration involved in inflammatory response | 0.0120 |
| GO:0045087 | innate immune response | 0.0191 |
| GO:0002526 | acute inflammatory response | 0.0193 |
| GO:0051493 | regulation of cytoskeleton organization | 0.0229 |
| GO:0051482 | positive regulation of cytosolic calcium ion concentration involved in phospholipase C-activating G-protein coupled signaling pathway | 0.0293 |
| GO:0048144 | fibroblast proliferation | 0.0311 |
| GO:0035924 | cellular response to vascular endothelial growth factor stimulus | 0.0311 |
| GO:0050832 | defense response to fungus | 0.0320 |

#### 3.2. Drug Repurposing for AD

Searching for the position of the AD-related genes in the combined drug-protein graph, we detected several cases not targeted by any drug. This is expected to some extent since the list of known drug-protein interactions covers a small set of proteins. However, by expanding

our search to second-degree neighbors, we are able to detect several drugs among them. In more detail, for the top 20 most important genes, only seven of them are targeted directly by a drug (Table 2). Including the second-order neighbors, we are able to find a drug in the subgraph for 17 of them. Some characteristic examples of proteins related to AD are shown in Figure 3. In Figure 3a, protein Q9H2W1 (encoded by gene Membrane-spanning 4-domains subfamily A member 6A (MS4A6A)) has no drugs among the neighbors, but it is connected with P13637 (ATP1A3), which is the target of two drugs, Ouabain (DB01092) and Rubidium chloride Rb-82 (DB09479). Similarly, in Figure 3b, protein Q13304 (Uracil nucleotide/cysteinyl leukotriene receptor (GPR17)) could be regulated by ATP (DB00171), which targets its neighbor P25098 (GRK2). Similarly, P02651 (Apolipoprotein C-I (APOC1)) could be potentially regulated via 14 drugs (Phenindione—DB00498, Dicoumarol—DB00266, Menadione—DB00170, Kappadione—DB09332, Navitoclax—DB12340, Zinc—DB01593, Phenprocoumon—DB00946, Acenocoumarol—DB01418, Warfarin—DB00682, Zinc acetate—DB14487, Copper—DB09130, Biotin—DB00121, Zinc chloride—DB14533, Zinc sulfate, unspecified form—DB14548) targeting five of its neighbors (P02647, Q9BQB6, Q92843, P02652, Q13085) as shown in Figure 3c.

**Table 2.** Important proteins and drugs targeting them.

| Protein | Gene | Importance | Drugs—Direct Interaction | Drugs—Immediate Interaction |
|---------|------|------------|--------------------------|------------------------------|
| P06702 | *S100A9* | 0.030 | 5 | 103 |
| Q9UBX3 | *SLC25A10* | 0.027 | 1 | 8 |
| P33552 | *CKS2* | 0.026 | 0 | 202 |
| P02654 | *APOC1* | 0.025 | 0 | 14 |
| Q9Y6R7 | *FCGBP* | 0.022 | 0 | 0 |
| Q9H2W1 | *MS4A6A* | 0.020 | 0 | 2 |
| O95183 | *VAMP5* | 0.020 | 0 | 58 |
| P30408 | *TM4SF1* | 0.020 | 0 | 18 |
| P28562 | *DUSP1* | 0.020 | 0 | 119 |
| Q9NY25 | *CLEC5A* | 0.019 | 0 | 0 |
| P13640 | *MT1G* | 0.018 | 2 | 72 |
| P01920 | *HLA-DQB1* | 0.017 | 1 | 10 |
| Q93091 | *RNASE6* | 0.017 | 0 | 0 |
| P61952 | *GNG11* | 0.017 | 0 | 2 |
| Q13304 | *GPR17* | 0.016 | 0 | 1 |
| O60356 | *NUPR1* | 0.016 | 0 | 18 |
| P05109 | *S100A8* | 0.016 | 6 | 213 |
| P08670 | *VIM* | 0.014 | 2 | 388 |
| P04792 | *HSPB1* | 0.014 | 3 | 538 |

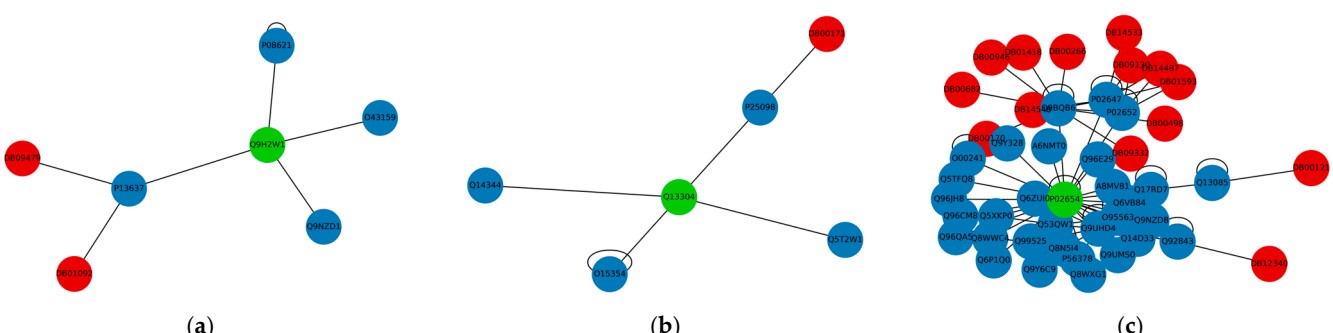

**Figure 3.** Drug-Protein interaction network around proteins of interest (green). (**a**) For Q9H2W1 (MS4A6A) (green), its neighbor P13637 (ATP1A3) is the target of two drugs, DB01092 and DB09479 (red). (**b**) Protein Q13304 interacts with P25098, which in turn is targeted by DB00171. (**c**) Protein P02651 is connected with five proteins targeted by 14 drugs.

## 4. Discussion

In this work, we created a static graph from drug-protein and protein-protein interactions, which can be used for any disease. Then, using scRNASeq data, we selected the most important genes with the help of machine learning methods. The Biological Process enrichment analysis (Table 1) confirmed that these genes play a role in neurodegenerative diseases. In detail, the most significant term "astrocyte development" is related to astrocytes, a sub-type of glial cells located in the brain and spinal cord, which have been linked with AD, since genes involved in AD are mainly expressed in glial cells [21,22]. The second most significant term, "positive regulation of intrinsic apoptotic signaling pathway", is related to cell death and has been suggested to play a role in neurodegenerative disease mechanisms [23]. In addition, it has been reported that in the early stages of AD, neutrophils accumulate in the blood [24], thus justifying the presence of the "neutrophil aggregation" term. "Sequestering of a zinc ion" is a process that takes place in neurons and it has been observed that related genes are under-expressed in AD patients [25]. Interestingly, "Peptidyl-cysteine S-nitrosylation" appears in the top terms, which is a chemical modification that involves the attachment of a nitric oxide (NO) group to the sulfur atom of a cysteine residue in a protein [26]. This modification, known as S-nitrosylation, can have significant effects on protein function and signaling pathways. Nitric oxide is a molecule with diverse roles in the body, including cellular signaling and regulation of blood vessels. Dysregulation refers to an abnormal or disrupted state of regulation. In this context, dysregulated protein S-nitrosylation suggests that the process of attaching nitric oxide to cysteine residues on proteins is not functioning properly. This dysregulation could lead to aberrant signaling and contribute to disease pathogenesis. Furthermore, the observation of dysregulated protein S-nitrosylation in AD implies that this chemical modification might play a role in the disease's pathophysiology [27,28]. It could potentially contribute to abnormal protein aggregation, neuroinflammation, oxidative stress, and synaptic dysfunction which are characteristic features of AD. Dysregulated S-nitrosylation could disrupt normal protein-protein interactions and cellular signaling processes, contributing to the cascade of events that drive disease progression.

Next, we investigated the role of the genes detected with high variable importance scores by the classification algorithm. Although these genes were selected solely in a data-driven way, they are indeed involved with neurodegenerative diseases. In the examples given in Figure 3, we focused only on some genes having no direct drug interaction, considering them as candidate therapeutic targets if they have a druggable interacting protein. In detail, the protein produced by the MS4A6A gene may be involved in signal transduction as a component of a multimeric receptor complex. Notably, it has been associated with aging and the onset of neurodegenerative diseases, with higher expression detected in AD tissues [29]. Additionally, according to the GWAS Catalog database (https://www.ebi.ac.uk/gwas, accessed on 5 August 2023), it is associated with an SNP related to the late onset of AD. The protein encoded by gene GPR17 is a dual specificity receptor for uracil nucleotides and cysteinyl leukotrienes (CysLTs), as well as signaling through G(i) and inhibiting adenylyl cyclase [30]. The GPR17 gene encodes a specific protein, and the statement provides insight into the protein's functional characteristics. The protein acts as a dual specificity receptor, meaning it has the capability to recognize and bind to two distinct types of molecules: uracil nucleotides and cysteinyl leukotrienes (CysLTs). This dual role suggests that the protein may play a role in signaling pathways associated with both types of molecules. Adenylyl cyclase is an enzyme responsible for producing cyclic AMP (cAMP), a secondary messenger molecule that participates in various cellular signaling pathways. The protein encoded by GPR17 inhibits adenylyl cyclase, which implies that it downregulates cAMP levels and consequently influences downstream cellular responses. This protein serves as a dual specificity receptor, recognizing both uracil nucleotides and CysLTs, and it exerts its effects by signaling through G(i) proteins to inhibit adenylyl cyclase. The protein's capacity to interact with multiple molecules and its involvement in inhibitory signaling pathways suggest its role in intricate cellular processes, which

could have implications for various physiological and pathological conditions. It has been supported that GPR17 is a sensor of brain damage and may play a role in both inducing neuronal death at early stages of injury and enabling repair response in later stages [31], while it has been shown that inhibition of it improves cognitive impairment [32]. Finally, the APOC1 protein plays a key role in high-density lipoprotein (HDL) and very low-density lipoprotein (VLDL) metabolism. Several polymorphisms in this gene have been associated with AD [33,34], while in the GWAS Catalog, APOC1 is related to a family history of AD.

The basis of network-based approaches is that proteins interact with each other to perform a function. Hence, both proteins of a PPI might have the same biological role, thus inhibiting one of them possibly can eliminate an undesired effect. Regarding the drugable neighbors of the selected proteins, all have been associated with AD. In detail, for MS4A6A, the drugable neighbor ATP1A3 is involved in various neurodegenerative diseases [35]; GPR17 interacts with GRK2 which has been found to be overexpressed in AD [36], while all five neighbors of APOC1 have been associated with AD (APOA1 [37], APOA2 [37], VKORC1 [38], BCL2L2 [39], and ACACA [40]).

Furthermore, regarding the drugs detected by our framework, several of them have been used in studies for the treatment of neurodegenerative diseases, although this is not the primary use for any of them, highlighting the significance of drug repurposing. Importantly, these drugs were selected indirectly based only on top-ranked genes/proteins identified by the classifier. Specifically, ATP (DB00171) is primarily used as nutritional supplementation and for treating dietary shortage or imbalance; however, it has been already used in a phase 2 study for AD (https://classic.clinicaltrials.gov/ct2/show/NCT02279511, accessed on 5 August 2023). Regarding the two drugs targeting a neighbor of the MS4A6A gene, both are related to heart condition treatments. Nonetheless, a study mentions Ouabain (DB01092) as a candidate for neurodegenerative diseases [41]. For APOC1, several drugs were found in its neighborhood, with the most promising case being Warfarin (DB00682), which is an anticoagulant drug normally used to prevent blood clot formation as well as migration. However, it has also been used in a phase 4 clinical trial for Cognitive Dysfunction and Dementia [42]. Other interesting cases include Biotin (DB00121) and Copper (DB09130), which have been used in clinical research for Amyotrophic Lateral Sclerosis (ALS). Biotin, also known as vitamin B7, is a water-soluble vitamin that plays a crucial role in various metabolic processes in the body. Biotin is involved in energy production, fatty acid synthesis, and the metabolism of amino acids and glucose. In the context of the statement, Biotin has been used in clinical research for ALS. This suggests that researchers are exploring the potential therapeutic effects of biotin supplementation in individuals with ALS. The rationale behind this research could be related to biotin's involvement in energy metabolism and its potential neuroprotective properties. Copper is an essential trace element that is vital for several physiological processes, including the formation of connective tissues, energy production, and the functioning of the nervous and immune systems. The statement indicates that copper has also been used in clinical research for ALS. This suggests that researchers are investigating whether copper supplementation or modulation could have beneficial effects in individuals with ALS. The potential connection between copper and ALS might stem from copper's role in various cellular processes and its influence on oxidative stress, which is implicated in neurodegenerative disorders like ALS.

However, there are some limitations in this work which can be the object of future research. First, the genes/proteins of interest are selected in a data-driven fashion from experimental data via Random Forest variable importance. Several other statistical and machine learning methods can be used to provide a ranking of genes with regard to their relationship with the disease or the user could even focus only on some genes of interest from the literature, which are confirmed to be related to a disease. Additionally, alternative sources can be used for drug-protein interactions, such as the the Therapeutic Target Database [43]. Finally, we note that some of the drugs with promising results have already been tested, but trials did not confirm their efficacy. Several reasons for this failure have been suggested [10], including the fact the AD mechanisms are not fully known,

thus reducing amyloid deposition might not be adequate. Additionally, some drugs might not be effective in the later stages of the disease. Furthermore, the small sample size and the neuropsychology metrics employed in Phase II trials might not be able to detect significant differences.

## 5. Conclusions

In the present study, we provide a framework, taking as input data of confirmed biological validity, such as refined PPIs and drug-protein interactions, as well as state-of-the-art gene expression data, i.e., scRNASeq. Then, using a machine learning approach, we select a subset of highly variable genes and extract an importance score. Combined with the graph, we are able to detect proteins of high importance without direct drug interactions, but with interacting proteins that are targets of known drugs. Based on the observations that interacting proteins usually show similar patterns in expression, it is possible to use these existing drugs to alleviate the effects of a different disease. Indeed, while further experimental studies remain imperative, it is worth emphasizing that the scope of our exploratory findings is confined to the recommendations furnished by Network Medicine, thereby leading to a concomitant reduction in the requisite temporal and financial commitments.

**Author Contributions:** Conceptualization, G.N.D., A.G.V., T.P.E. and M.G.K.; methodology, G.N.D., A.G.V., T.P.E., M.G.K. and P.V.; validation, P.V.; formal analysis, G.N.D. and P.V.; data curation, G.N.D., A.G.V., T.P.E. and M.G.K.; writing—original draft preparation, G.N.D., A.G.V., T.P.E. and M.G.K.; writing—review and editing, G.N.D., A.G.V., T.P.E., M.G.K. and P.V.; funding acquisition, T.P.E. All authors have read and agreed to the published version of the manuscript.

**Funding:** This research is funded by the European Union and Greece (Partnership Agreement for the Development Framework 2014–2020) under the Regional Operational Programme Ionian Islands 2014–2020, project title: "Study of drug protocols with biomarkers that define the evolution of non-genetic neurodegenerative diseases—NEUROPHARMA", project number: 5016117.

**Institutional Review Board Statement:** Not applicable.

**Informed Consent Statement:** Not applicable.

**Data Availability Statement:** Not applicable.

**Conflicts of Interest:** The authors declare no conflict of interest.

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
