# Peer review of "Drug and Protein Interaction Network Construction for Drug Repurposing in Alzheimer’s Disease"

_futurepharmacol, doi:10.3390/futurepharmacol3040045_

Round 1

Reviewer 1 Report

This manuscript explains an interesting approach in the search of new indications for drugs already in use.

The initiative is quite valuable and could be of importance in the search of new treatments.

There are some things to improve in the manuscript.

Introduction: The last paragraph is more suitable for the Material and Methods section

Results: In this section the authors do not give the names of the drugs candidates to repurposing, even though they do in some cases in the discussion. It would be useful to have identified the drugs that have been selected in the study.

Discussion: Some of the drugs that have promising results, have been already tested for AD and have not given good outcomes. Can the authors comment this aspect?

Author Response

This manuscript explains an interesting approach in the search of new indications for drugs already in use.

The initiative is quite valuable and could be of importance in the search of new treatments.

There are some things to improve in the manuscript.

Reply: We would like to thank Reviewer for his/her kind words and helpful comments. We addressed below point-by-point the suggested comments.

Introduction: The last paragraph is more suitable for the Material and Methods section

Reply: Thank you for the suggestion. We have restructured the last introduction paragraph, to highlight what was done and not how it was done, with moving some details to methods.

In this work, our aim is to detect potential drug targets for AD based on known drug-proteins and protein-protein interactions (PPIs). Focusing on proteins that play a significant role in AD but without known drugs, the key idea is to detect drugable neighbors of these proteins. Towards this aim, we created a drug-protein graph from high confidence PPI and drug-protein interaction data. Then, a scRNASeq dataset on AD was used as input to a classification method in order to detect genes that their ex-pression presents different patterns between AD and control conditions. Then, for the corresponding proteins of the top ranked genes, we searched in the graph if they were targeted by a drug or not. In the latter case, we searched for drugs targeting another protein that is linked with a direct PPI. Further analyzing these drugs, we found that in several cases, although their primary aim is not AD, they have the potential to be repurposed for this disease. 

Results: In this section the authors do not give the names of the drugs candidates to repurposing, even though they do in some cases in the discussion. It would be useful to have identified the drugs that have been selected in the study.

Reply: In the revised manuscript, we provide the names of the drugs along with the code.

Discussion: Some of the drugs that have promising results, have been already tested for AD and have not given good outcomes. Can the authors comment this aspect?

Reply: Thank you for the comment, this point has been included in the discussion.

Finally, we note that some of the drugs with promising results have already been test-ed, but trials did not confirm their efficacy. Several reasons for this failure have been suggested [10], including the fact the AD mechanisms are not fully known, thus reducing amyloid deposition might not be adequate. Additionally, some drugs might not be effective on later stages of the disease. Furthermore, the small sample size and the neuropsychology metrics employed in Phase II trials might not be able to detect significant differences.

Reviewer 2 Report

This manuscript, entitled 'Drug and protein interaction network construction for drug 2 repurposing in Alzheimer's Disease', is an article that addresses a critical issue in Alzheimer's disease research, namely the identification of candidate therapies through drug repurposing. The integration of protein-protein interaction (PPI) networks and drug-target interactions is a novel and promising approach. The methodology, which involves assigning edge weights based on co-expression and estimating differential gene expression, is well described and seems scientifically sound. Overall, this study makes a valuable contribution to the field of Alzheimer's disease research and drug repurposing and paves the way for more precise Network Medicine strategies. Although the general approach of the study is promising, there are some areas where improvements could be made.

1. The introduction should report on examples of repurposed drugs in AD 10.1038/s41582-020-0397-4 : as example in the field of metallodrugs several interesting studies concern anticancer cysplatin derivative as modulators of amyloid aggregation 10.3390/ijms20040829 10.1021/acs.inorgchem.1c03540

2.  Could you elaborate on the machine learning methods used for gene selection? What specific features or variables were considered in the Random Forest variable importance analysis, and how were these features related to neurodegenerative diseases? Additionally, were any validation techniques employed to assess the robustness of the gene selection process?

3. The methodology for assigning weights to edges based on co-expression and estimating differential gene expression should be explained in more depth. Readers would benefit from a clearer understanding of the mathematical and statistical techniques used.

4.  When discussing the significance of drug repurposing, could you explain why these specific drugs were chosen as potential candidates for neurodegenerative disease treatment and provide insights into the rationale behind their selection?

5. Please, when discussing about the Peptidyl-cysteine S-nitrosylation (page 7, lines 134-248), especially when discussing its implication in the pathology of AD,  it would be better add some reference.

6. Regarding the identified genes with no direct drug interaction but potential druggable interacting proteins, such as MS4A6A and GPR17, can you provide insights into their functional significance within neurodegenerative disease pathways? How might targeting these genes indirectly through their druggable interacting partners impact disease mechanisms? Even if the manuscript is quite methodological and theoretical the authors should pay a little attention to the experimental

 Minor revisions

1) Improve the resolution of Figure 2a and figure 3.

2) Pay attention to the punctuation and the English form.

Careful attention to the punctuation and the building of sentences

Author Response

This manuscript, entitled 'Drug and protein interaction network construction for drug 2 repurposing in Alzheimer's Disease', is an article that addresses a critical issue in Alzheimer's disease research, namely the identification of candidate therapies through drug repurposing. The integration of protein-protein interaction (PPI) networks and drug-target interactions is a novel and promising approach. The methodology, which involves assigning edge weights based on co-expression and estimating differential gene expression, is well described and seems scientifically sound. Overall, this study makes a valuable contribution to the field of Alzheimer's disease research and drug repurposing and paves the way for more precise Network Medicine strategies. Although the general approach of the study is promising, there are some areas where improvements could be made.

Reply:  We would like to thank Reviewer for his/her constructive and thoughtful comments that assisted us in preparing a much-improved revision of the manuscript. We addressed below point-by-point the suggested comments.

  1. The introduction should report on examples of repurposed drugs in AD1038/s41582-020-0397-4 : as example in the field of metallodrugs several interesting studies concern anticancer cysplatin derivative as modulators of amyloid aggregation 10.3390/ijms20040829 10.1021/acs.inorgchem.1c03540

Reply: Thank you for the comment. We have expanded the introduction to include more references to drug repurposing, including the mentioned ones.

Researchers can save time using already-approved drugs, as they can perform directly phase II clinical trials. Several promising compounds are reported that are prioritized to be used in clinical trials with AD patients [10]. For example, several interesting studies in the field of metallodrugs consider anticancer cisplatin derivative as modulator of amyloid aggregation [11,12].

  1. Could you elaborate on the machine learning methods used for gene selection? What specific features or variables were considered in the Random Forest variable importance analysis, and how were these features related to neurodegenerative diseases? Additionally, were any validation techniques employed to assess the robustness of the gene selection process?

Reply: The relevant content has been extended to cleat the methods we followed. Each gene was considered as variable. Variable importance as calculated by Random Forest algorithm is a widely used approach to select variables. It is considered robust, since it is estimated as the average over a large number of trees. The robustness is enhanced via randomness:  a) each tree receives a random subsample of data and b) in each node some variables are selected randomly to define the best split. Accordingly, we proceeded in the revised manuscript:

Results of Random Forest are considered more robust especially compared to single tree classification due to randomness: a) each tree receives a random subsample of data and b) in each node some variables are selected randomly to define the best split. Variable importance measurement in the context of Random Forest refers to assessing the significance of individual input features (here genes) in contributing to the model's overall predictive performance. We used the normalized Gini Importance, which calculates the total reduction in the Gini impurity (a measure of node impurity in decision trees) achieved by a particular feature over all trees in the Random Forest. Higher values of the variable importance show higher reduction, indicating that the feature is considered more important.

  1. The methodology for assigning weights to edges based on co-expression and estimating differential gene expression should be explained in more depth. Readers would benefit from a clearer understanding of the mathematical and statistical techniques used.

Reply: Thank you for this point. There is a misunderstanding, since in this paper there was no weight assigned on edges. The gene expression was only used in classifier using genes as features and conditions as target, in order to get feature (gene) importance, i.e., the ability of gene expression to distinguish the two conditions. The text was improved to clear this point.

The scRNASeq dataset was used to detect the most significant nodes related with AD and obtain a ranking of their ability to distinguish health and disease conditions. In detail, the Random Forest method was employed to classify the cells as control or disease, based on gene expression, i.e., the genes were considered as features and the cell samples as observations.

  1. When discussing the significance of drug repurposing, could you explain why these specific drugs were chosen as potential candidates for neurodegenerative disease treatment and provide insights into the rationale behind their selection?

Reply: Thank you for your comment. To showcase our method, we selected some proteins with high importance (as shown in Table 1) but without any known direct drugs targeting them. Then, we analyzed the immediate drug connections and further discussed their potential relationship with AD. Thus, these drugs were selected indirectly, via the targeted proteins and their neighbors. The text was amended to emphasize on this.

Next, we investigated the role of the genes detected with high variable importance score by the classification algorithm. Although these gene were selected solely in a data-driven way, they are indeed involved with neurodegenerative diseases.

We remind that these drugs were selected indirectly, based only on top-ranked genes/proteins identified by the classifier.

  1. Please, when discussing about the Peptidyl-cysteine S-nitrosylation (page 7, lines 134-248), especially when discussing its implication in the pathology of AD, it would be better add some reference.

 Reply: Thank you for the suggestion, some references were added.

  1. Regarding the identified genes with no direct drug interaction but potential druggable interacting proteins, such as MS4A6A and GPR17, can you provide insights into their functional significance within neurodegenerative disease pathways? How might targeting these genes indirectly through their druggable interacting partners impact disease mechanisms? Even if the manuscript is quite methodological and theoretical the authors should pay a little attention to the experimental

Reply: These genes were selected in a data-driven way, as most differentially expressed. In discussion, the possible relation of these genes with AD is analyzed. Furthemore, a paragraph was added to provide evidence that their neighbors are also involved in neurodegenerative diseases. Of note, MSA4A6A is not yet mapped in any pathways and GPR17 is found in unrelated ones (based on KEGG database).

The basis of network-based approaches is that proteins interact with each other to perform some function. Hence, both proteins of a PPI might have the same biological role, thus inhibiting one of them possibly can eliminate some undesired effect. Regard-ing the drugable neighbors of the selected proteins, all have been associated with AD. In detail, for MS4A6A, the drugable neighbor ATP1A3 is involved in various neuro-degenerative diseases [35]; GPR17 interacts with GRK2 which has been found to be overexpressed in AD [36], while all five neighbors of APOC1 have been associated with AD (APOA1 [37], APOA2 [37], VKORC1 [38], BCL2L2 [39] and ACACA [40]). 

Minor revisions

  • Improve the resolution of Figure 2a and figure 3.

Reply: Higher resolution images were provided.

2) Pay attention to the punctuation and the English form.

Reply: We have proof-read the manuscript and made several corrections.

Round 2

Reviewer 2 Report

The authors addressed all my raised isses

It is clear in my opinion